# Vitamin A, D, E, and K as Matrix Metalloproteinase-2/9 Regulators That Affect Expression and Enzymatic Activity

**DOI:** 10.3390/ijms242317038

**Published:** 2023-12-01

**Authors:** Ha Vy Thi Vo, Yen Thi Nguyen, Namdoo Kim, Hyuck Jin Lee

**Affiliations:** 1Department of Chemistry Education, Kongju National University, Gongju 32588, Republic of Korea; vothihavy11102000@gmail.com; 2Department of Chemistry, Kongju National University, Gongju 32588, Republic of Korea; hoangyen666.yh@gmail.com; 3Kongju National University Institute of Science Education, Kongju National University, Gongju 32588, Republic of Korea; 4Kongju National University’s Physical Fitness for Health Research Lab (KNUPFHR), Kongju National University, Gongju 32588, Republic of Korea

**Keywords:** fat-soluble vitamin, matrix metalloproteinases, cancer

## Abstract

Fat-soluble vitamins (vitamin A, D, E, and K) assume a pivotal role in maintaining human homeostasis by virtue of their enzymatic functions. The daily inclusion of these vitamins is imperative to the upkeep of various physiological processes including vision, bone health, immunity, and protection against oxidative stress. Current research highlights fat-soluble vitamins as potential therapeutics for human diseases, especially cancer. Fat-soluble vitamins exert their therapeutic effects through multiple pathways, including regulation of matrix metalloproteinases’ (MMPs) expression and enzymatic activity. As MMPs have been reported to be involved in the pathology of various diseases, such as cancers, cardiovascular diseases, and neurological disorders, regulating the expression and/or activity of MMPs could be considered as a potent therapeutic strategy. Here, we summarize the properties of fat-soluble vitamins and their potential as promising candidates capable of effectively modulating MMPs through multiple pathways to treat human diseases.

## 1. Introduction

### 1.1. Fat-Soluble Vitamins (A, D, E, K) 

Fat-soluble vitamins are a group of vitamins that can dissolve in fats and be absorbed along with dietary fats in the human digestive tract. There are four primary fat-soluble vitamins, A, D, E, and K, and each has its own unique dietary sources. Unlike water-soluble vitamins (e.g., vitamin B and C), which are not stored in large quantities in the human body and are excreted in the urine when consumed in excess, fat-soluble vitamins can be stored in fatty tissues and liver [1,2]. This indicates that the body can utilize these stored vitamins during periods of inadequate dietary intake. Fat-soluble vitamins are essential for maintaining physiological equilibrium within the human body, encompassing the maintenance of bone health, vision, cell growth, immune functions, and acting as a powerful antioxidant. Additionally, they act as cofactors or regulators for various enzymes and biochemical processes in the body [2]. Vitamin A is necessary for regulating visual cycles and the expression of various genes involved in cell growth, differentiation, and development [3,4]. Vitamin D is involved in calcium homeostasis, immune system regulating, cell proliferation and differentiation, and cytokine production [5,6]. Vitamin E acts primarily as an antioxidant, protecting cellular components (i.e., DNA, proteins, lipid, and enzyme like ribonucleases) from oxidative damage [7,8,9]. Vitamin K serves as a cofactor for enzymes involved in the synthesis of clotting factors in the liver. It also plays a role in bone metabolism by modifying proteins involved in calcium binding and bone mineralization [10]. 

Contemporary studies underscore the potential of fat-soluble vitamins as promising therapeutic agents for the treatment of human diseases, with a particular focus on their efficacy in cancer treatment. They could ameliorate human diseases through various mechanistic pathways, with a notable emphasis on transcriptional regulation and signaling modulation. Vitamin A and D act as cofactors in transcriptional regulation [11,12,13]. In contrast, vitamin E and K could not serve as traditional cofactors. Vitamin E may influence signaling pathways by affecting the activity of kinases and phosphatases, while vitamin K is generally associated with anti-inflammatory effects by modulating posttranslational modifications [14,15,16]. Specifically, fat-soluble vitamins have been shown to regulate the production of matrix metalloproteinases (MMPs) which are involved in the pathogenesis of various diseases, including cancers, cardiovascular diseases, and neurological disorders [17,18,19,20,21].

### 1.2. Matrix Metalloproteinases-2 and -9

MMPs are a family of zinc-dependent proteolytic enzymes that have different substrates, but which share the similar structural characteristics. They are involved in the breakdown and remodeling of the extracellular matrix (ECM) in various tissues of the body. MMPs are produced by various cell types, including fibroblasts, immune cells, and endothelial cells (Table 1) [22,23]. MMPs could facilitate tissue remodeling during embryogenesis, tissue repair, and angiogenesis [24,25]. The dysregulation and/or overexpression of MMPs results in the progression of various diseases including cancer invasion and metastasis, arthritis, brain degenerative diseases, cardiovascular diseases, and tissue fibrosis [22,26,27]. Among MMPs, MMP-2 and MMP-9 are particularly prominent due to their involvement in a wide range of human diseases. MMP-2 (gelatinase A, type IV collagenase) is expressed ubiquitously as a 72 kDa proenzyme and is heavily glycosylated [28,29]. The expression of MMP-2 is consistent and is not significantly increased by proinflammatory triggers, as its gene lacks certain binding sites for proinflammatory transcription factors [28]. MMP-9 (gelatinase B, type IV collagenase) is expressed as a 92 kDa proenzyme and requires activation for its proteolytic activity [30,31]. The transcription of MMP-9 is regulated by transcription factors such as nuclear factor kappa-B (NF-κB), specificity protein 1 (SP1), and activator protein 1 (AP1), which respond significantly to inflammatory stimuli [32].

The activity of MMP-2 and MMP-9 is tightly regulated by the endogenous tissue inhibitors of metalloproteinases (TIMPs), through negative feedback acting on activation of MMPs [33,34]. TIMP-2, TIMP-3, and TIMP-4 can regulate MMP-2 activity, while TIMP-1 and TIMP-3 can regulate that of MMP-9 [22,35]. Under normal conditions, ECM homeostasis is interdependent between MMP and TIMP activities. Under pathologic conditions, however, MMPs are overexpressed or under controlled by TIMPs, resulting in the dysregulation of tissue remodeling which can cause a variety of diseases [36]. MMP-2 and MMP-9 play a key role in cancer progression and metastasis by degrading ECM thus, they are important predictive factors for various cancers. Overexpression of MMP-2 and MMP-9 was linked to poor prognosis in oral cancer [37], lung cancer [38], breast cancer [39], retinoblastoma [40], bladder cancer [41], and ovarian epithelial cancer [42]. Activated MMP-2 and MMP-9 contribute to cell invasion by breaking down collagen type IV in the basement membrane [43]. Tumor tissue analysis revealed significant levels of MMP-2 and MMP-9 expression and their active forms [44]. The excessive presence of MMP-2 triggers the activation of p38 MAPK/ML/SHP27 signaling, leading to actin polymerization that supports cell migration [45]. Moreover, MMP-2 and MMP-9 participate in angiogenesis, thereby enhancing tumor growth and development [46].
ijms-24-17038-t001_Table 1Table 1Expression and activation of MMP-2 and MMP-9 in different cell types.MMPLocationSubstrateCell TypesBiological FunctionRef.MMP-2ECMType IV Collagen, Gelatin, Fibronectin,Laminin, Aggrecan,Versican, Elastin-Matrix remodeling,angiogenesis,inflammation,invasion and metastasis[28,47]Cytoplasmα-actinin, Tn1, Titin, MLC-1, Troponin ICardiomyocytesContractile dysfunction[48,49]GSK-3βCardiomyoblastApoptosis[50]TalinPlateletsAggregation[51,52]MitochondriaHsp60, Cx43Retinal endothelial cellsApoptosis[53,54,55,56]IκB-αMyoblastic cellsNucleusPARP1, XRCC1NeuronsMMP-9ECMType IV Collagen,Gelatin, Fibronectin,Laminin, Aggrecan,Versican, Nidogen,Tenascin, CollagenX,Collagen III, Elastin-Matrix remodeling,angiogenesis,inflammation,invasion and metastasis[28,47]CytoplasmAMPKαLeukocytesInnate immunity[57,58]MHCCardiomyocytesMitochondriaCx43, Hsp60, Hsp70Cardiomyocytes, Retinal cellsApoptosis[59,60]NucleusPARP1, XRCC1NeuronsApoptosis[55,61]Histone H3, Citrate synthaseOsteoclast, Cardiomyocytes

MMP-2 and MMP-9 are also associated with cardiovascular diseases (e.g., atherosclerosis, myocardial infarction, and heart failure) [62]. They contribute to plaque instability and rupture in atherosclerotic lesions, potentially leading to heart attacks or strokes [63]. MMP-2 and MMP-9 are linked to neuroinflammation and neurodegenerative diseases such as Alzheimer’s disease (AD) and multiple sclerosis as well. Enhancement of MMP-2 could form neurofibrillary tangles in neurons suggesting that MMP-2 could stimulate tau formation. Also, upregulation of MMP-9 levels and/or activity could contribute to the breakdown of the blood–brain barrier (BBB) and promote inflammation [64,65]. Moreover, MMP-2 and MMP-9 are involved in inflammatory joint diseases, including rheumatoid arthritis, through degradation of cartilage and joint tissues. Particularly, the excessive activity of MMP-9 can contribute to joint damage and functional impairment [66]. Therefore, inhibiting the enzymatic activity of MMP-2 and MMP-9 has been explored as a potential target for various human diseases.

## 2. Vitamin A

The term vitamin A indicates any compound possessing the biological activity of retinol (e.g., retinol, retinyl esters, retinal, retinoid acid, and oxidated and conjugated forms of both retinol and retinal) [67]. Vitamin A is found in variety foods and exists in two primary forms: (i) retinol and retinyl esters, which are the active form of vitamin A and found in animal sources, and (ii) the carotenoids who function as provitamin A and are found in vegetables [67,68]. Carotenoids can be metabolized to retinal and then to retinols [69]. Vitamin A is essential to maintain various physiological functions in the human body, including vision, immune system, and cell communication [70,71].

### 2.1. Transportation and Metabolism of Vitamin A

Different forms of vitamin A are solubilized into micelles in the intestinal lumen and absorbed by duodenal mucosal cells. As presented in Figure 1, carotenoids are converted to retinal and then reduced to retinol [72]. Retinol is then esterified with long-chain fatty acids into retinyl esters. Retinyl esters and intact carotenoids bind to the lipids (e.g., cholesterol, cholesterol esters, and triglycerides) to form chylomicrons. These chylomicrons are transferred through the lymphatics to the bloodstream [73]. In the bloodstream, chylomicrons are hydrolyzed into chylomicron remnants by lipoprotein lipase and apolipoprotein E. The chylomicron remnants can deliver retinyl esters directly to target cells and/or be taken up by hepatocytes [74]. In hepatocytes, retinyl esters are hydrolyzed to retinol. Retinol is then stored in the liver or transported to the blood stream when it is necessary for biological functions, such as regulation of the visual cycle, cell growth and differentiation. The liver is the main storage site of vitamin A (approximately 70% to 80% of total body stores) and is a central player in the homeostasis of vitamin A [75]. Smaller amounts of retinyl esters, as well as carotenoids are also carried by chylomicrons and chylomicron remnants to extrahepatic tissues for use and storage [76]. The hydrolyzed retinol in stellate cells is believed to be transported back to hepatocytes, where it binds to retinol binding protein (RBP), forming a retinol–RBP complex that enters circulation [77]. This complex combines with transthyretin and effectively prevents the complex from being cleared by kidney [78].

### 2.2. Cell Intake and Intracellular Activities

Different forms of vitamin A, such as retinol–RBP complex and retinyl esters/carotenoids–chylomicrons, are taken up by peripheral cells in the plasma. There, carotenoids and retinyl esters are metabolized to retinol [69]. Retinol is then oxidized to all-trans retinal, and then oxidized to all-trans retinoic acid (ATRA) which is the most hormonally active retinoid and is not reconverted to retinol in biological systems [67,79]. The cell metabolism and various signaling pathways of retinoids require many different binding proteins and receptors. The genomic effects of retinoids are mediated primarily by two families of nuclear hormone receptors: retinoic acid receptors (RARs) and retinoid X receptors (RXRs). ATRA binds to RARs and RXRs, which act as ligand-activated transcription factors. These receptor–ATRA complexes bind to retinoic acid response elements (RAREs) in the promoter regions of target genes, leading to either activation or repression of gene transcription [80]. This activity regulates the transcription of a large number of genes, primarily involved in cellular differentiation, proliferation, and apoptosis, such as Insulin-like Growth Factor-Binding Protein 3 (IGFBP-3), BCL-2, HOXA and HOXB genes [81]. Additionally, ATRA also binds to retinoid-related orphan receptors (RORs) to initiate transcription by activating specific ROR response elements (ROREs) in DNA [82]. Retinoids can have rapid nongenomic/nonclassical actions, by inducing a rapid phosphorylation of the cAMP response element binding protein (CREB). This process causes CREB to relocate to the nucleus and activates genes containing cAMP response elements (CREs) in their promoters [83]. Thus, ATRA could be considered as an effective signal molecule for regulating gene expression. Numerous genes can be regulated by retinoids either directly or indirectly, including genes related to MMPs, such as stromelysin-1, collagenases, and gelatinases [81,84]. The actual number of retinoid-regulated genes, and regulatory pathways can vary depending on cell type, tissue, developmental stage, and other contextual factors [11].

### 2.3. Homeostasis of Vitamin A and Related Diseases

Vitamin A is one of the most versatile vitamins in the human body. It plays crucial roles in various essential physiological processes (Table 2). The majority of the biological functions are not directly performed by retinol itself but instead by its active metabolites (retinal and retinoic acids). They are involved in different activities including vision, immunity, cell differentiation, embryological development, cellular differentiation and proliferation and antioxidant functions [4,85].

For vision, rhodopsin, the light-sensitive pigment found in the eye rods, is formed through the binding of 11-*cis*-retinal to opsin. Light absorption triggers a series of reactions to all-trans-retinal and opsin, transmitting a visual signal. Night blindness can result from vitamin A deficiency, reducing 11-*cis*-retinal and rhodopsin levels, leading to a weakened response to low light at night [86]. For the immune system, vitamin A is essential for the maintenance of skin health and mucous membranes, acting as a barrier against infections. It is involved in the production of white blood cells and the activation of the immune response by suppressing the production of proinflammatory cytokines [87]. Vitamin A deficiency can increase the risk of infection [87]. The role of Vitamin A in regulating growth through cell proliferation and differentiation has been acknowledged as well [102,103]. As mentioned above, retinoic acids regulate the transcription of various genes which involve cellular functions [104,105]. In addition, new biological functions related to insulin resistance, lipid metabolism, energy balance, and redox signaling have been described [97]. Due to the diverse activities, vitamin A has been studied for its therapeutic effects against various human diseases, such as infectious diseases [87], APL [88,89,90], osteoporosis [91,92], obesity and insulin resistance [96,97] and cancer [17,18,93,94,95].

In the absence of vitamin A, (1) proper stem cell differentiation does not occur; (2) growth and development of embryos are altered; (3) epithelial cellular development is deficient and the barrier to infection is decreased; (4) cells involved in innate and acquired immune function are decreased; (5) xerophthalmia develops because of abnormalities in corneal and conjunctiva development; and (6) normal bone growth and tooth development do not occur [70,71,106]. Deficiency of vitamin A before and during pregnancy is believed to be associated with an increased risk of congenital malformations and impaired vascularized development [98,99].

### 2.4. Vitamin A in Cancer Treatment and MMP Regulation

Different forms of vitamin A have been studied for their role in cancer treatment for many years. ATRA has been approved by the FDA in 1995 for the treatment of APL, an aggressive blood cancer. It is believed that vitamin A is related to the production and activity of MMP-2 and MMP-9 in cancer progression, metastasis, and other human diseases [107]. Vitamin A could selectively regulate the expression and/or activity of MMP-2 and MMP-9 within distinct cellular contexts. This selectivity is attributed to the different roles of MMPs in various tissues and cell types, which lead to differences in RA signaling. MMP-2 and MMP-9 production was upregulated upon treatment of RA in murine dendritic cells [108], leukemic cells [90], mesenchymal stem cells [109] and red deer antler stem cells (ASCs) [110]. By contrast, the downregulation of MMP-2 and MMP-9 expression with the addition of vitamin A was found in various cancer cell lines including breast cancer [111], lung cancer [100], glioblastoma [93], and chondrosarcoma [112].

Regarding cancer immunology, cancer exerts systemic impact on immune cell function via various mechanisms. Notably, a key mechanism used by tumors to suppress the immune response is the sequestration of dendritic cells (DCs) within tumor tissues. This sequestration inhibits the mobility of these immune cells, leading to immunosuppression [113,114]. MMPs have been shown to be essential for DC movement through basement membranes and the ECM [115]. Therefore, improvement of DCs’ mobility via enhancing the production of MMPs, especially MMP-9, was believed to be a potential strategy to counteract the immunosuppression observed in tumors [115]. MMP-2 and MMP-9 mRNA expression and production were increased by five-fold (along with the decrease in the production of their inhibitors, TIMPs) upon treatment of ATRA in DCs [108]. The balance of MMPs and their inhibitors was suggested to be beneficial for DC trafficking in the tumor milieu, improving immune responses in cancer patients. This finding was confirmed by later research that showed the upregulation of MMP-2 and MMP-9 expression and activity upon treatment with ATRA, investigated in bone marrow-derived mesenchymal stem cells [109].

On the other hand, MMP-2 and MMP-9 expression could be decreased in the presence of vitamin A. β-carotene can inhibit neuroblastoma cell invasion via different pathways (i.e., suppressing the expression and activity of MMP-2) [17]. In neuroblastoma cells (SK-N-BE(2)-C cells), a decrease in MMP-2 expression and activity was observed with the treatment of β-carotene. Also, β-carotene could downregulate the expression of HIF-1α, a factor that activates the transcription of many genes, including vascular endothelial growth factor (VEGF) which is involved in the upregulation of MMP-2 and MMP-9 expression and activity. The decrease in MMP-2 and MMP-9 expression and activity was also observed in Lewis lung carcinoma cells upon addition of β-carotene [95]. In gastric cancer cells, AGS and SGC-7901, the level of both MMP-2 and MMP-9 was significantly decreased upon β-cryptoxanthin (a pro-vitamin A, β-carotene subclass) addition [101]. Expression of MMP-9 and NF-κB were also reported to be decreased in colorectal cancer cells and paclitaxel-resistant colorectal cancer cell lines (HCT116, LoVo and CT26) with ATRA treatment [94].

In addition, ATRA could suppress colorectal cancer cells’ (RKO) migration via downregulating ERK/MAPK [18]. An earlier study also proposed that RA suppressed the expression of MMP-2 in rat lung fibroblasts (LFs) through decreasing Jun N-terminal kinase (JNK) and p38 activation in hyperoxia. Moreover, RA could inhibit MMP-2 secretion in T-98G cells and decrease SF2 levels in HL-60 cells [116]. SF2 is a proto-oncogene which is involved in the alternative splicing of Mcl-1, a protein that inhibits apoptosis, and plays a role in regulating VEGF. Considering that VEGF is believed to be implicated in the regulation of MMP expression [117,118], it is plausible that RA treatment could lead to a reduction in MMP expression.

## 3. Vitamin D

Vitamin D is another fat-soluble vitamin that can be obtained through two major sources; natural dietary sources and synthesis by skin when it is exposed to UVB rays from sunlight [119]. The two main forms of vitamin D, crucial for human body, are ergocalciferol (vitamin D_2_) and cholecalciferol (vitamin D_3_). Vitamin D_2_ can be obtained from plant-based sources, while vitamin D_3_ is mainly synthesized by skin or gained from animal sources [119,120]. The structures of vitamins D_2_ and D_3_ differ in the side chain; D_2_ contains a double bond (C_22–23_) and an additional methyl group on C_24_ (Figure 2). The production of vitamin D_3_ in the epidermis is initiated by the action of UVB rays on 7-dehydrocholesterol. The UVB rays break the B ring of the cholesterol structure to form previtamin D_3_, then undergoes a thermal induced rearrangement to form vitamin D_3_ [121].

### 3.1. Transportation and Metabolism of Vitamin Ds

Vitamin D_2_ and D_3_ obtained from food are soluble in micelles in the small intestine, where they are mainly absorbed by the apical membrane of enterocytes [122]. After absorption, they bind to other lipids to form chylomicrons, which can enter the bloodstream (Figure 3). In the bloodstream, chylomicrons–D_2_/D_3_ and cutaneous vitamin D_3_ bind to vitamin D binding protein (DBP) and are transported to various tissues and organs, including the liver [123,124]. Vitamin D_2_ and D_3_ are metabolized to 25(OH)D_2_ and 25(OH)D_3,_ respectively, through the actions of several cytochrome P450s (CYP) exhibiting 25-hydroxylase activity, such as CYP2R1 and CYP27A1. The major and the most stable vitamin D metabolites are 25(OH)Ds with a serum circulation half-life of 15 days [120]. This hydroxylation takes place primarily in the liver, but also in other tissues (i.e., skin, adipose tissue, immune cells and osteoblasts). These vitamin D metabolites, 25(OH)Ds, are stored in the cytoplasm of hepatocytes in a limited amount at about 10 nmol/kg body weight under normal conditions. Additionally, 25(OH)Ds circulate in the bloodstream, in a bound form with DBP at a concentration of 45.25 nmol/kg. The 25(OH)Ds are also taken up by fat tissues and stored in fat cells at 5 nmol/kg [125]. The 25(OH)Ds are also transported to the kidneys where they are further metabolized to the more biologically active forms, 1,25(OH)_2_Ds (Calcitriol). This process is achieved by the enzyme 25OHD-1α hydroxylase (CYP27B1) [126,127]. Although CYP27B1 is mostly expressed in epidermal keratinocytes, it is found in the renal tubules of the kidney, as well as in various other tissues and organs, such as the brain, placenta, testes, intestine, lung, breast, macrophages, lymphocytes, parathyroid gland, osteoblasts, and chondrocytes [126,127,128,129]. Under normal conditions, the kidney is typically recognized as the primary contributor to circulate 1,25(OH)_2_Ds. However, in certain pathological situations, extra renal CYP27B1 activities in other tissues can play a role in generating 1,25(OH)_2_Ds, leading to elevated levels of this active vitamin D and calcium [126]. In kidney, 25(OH)Ds are also metabolized by 25OHD-24 hydroxylase (CYP24A1) into 24,25(OH)_2_Ds, which are the second most important and inactive metabolites of 25(OH)Ds. CYP24A1 also catabolizes 1,25(OH)_2_Ds to their inactive forms (1,24,25(OH)_3_D or 1,23,25(OH)_3_D) in cases where vitamin D is overactive [130,131]. Thus, CYP24A1 plays a crucial role in regulating the body level of 1,25(OH)_2_Ds.

The 25(OH)Ds and 1,25(OH)_2_Ds are circulated in the bloodstream in a primarily bound form with DBP and are subsequently transported to target tissues. In pathological conditions such as liver disease and nephrotic syndrome, reduced levels of DBP and albumin can lead to lower total levels of 25(OH)Ds and 1,25(OH)_2_Ds, but the concentrations of these labile metabolites are not affected [132]. Similarly, DBP levels are reduced during acute illness, potentially obscuring the interpretation of total 25(OH)D levels [133]. Thus, DBP–vitamin D complexes play an important role in transporting vitamin D and its metabolites to target organs and tissues. However, the level of 1,25(OH)_2_D in tissues were reported to be unchanged without DBP [134,135,136]. This suggests that DBP-unbound vitamin D metabolites may be more crucial than the bound-form and can be taken up immediately into cells because DBP is not available to most cells [127,128,129].

### 3.2. Cell Uptake and Intracellular Functions of Vitamin D

After entering target cells, 1,25(OH)_2_Ds act as transcriptional regulators by binding with nuclear vitamin D receptors (VDRn) in the ligand binding region. The VDRn then pairs with the RXR and binds to the vitamin D response element in the promoter region of the gene. Coactivator proteins, including vitamin D receptor-interacting protein (DRIP) and steroid receptor coactivator (SRC) could form the complex with VDRn/RXR. This complex facilitates the transcription of the gene to produce mRNA, which is then translated into the corresponding protein. More than 200 genes (almost 3% of human genome) are up or downregulated by vitamin D [13]. Flanking gene sequences and tissue specific factors influence the regulation of gene expression by 1,25(OH)_2_Ds [137].

Moreover, 1,25(OH)_2_Ds are also involved in non-genomic actions via their binding forms with a distinct putative plasma membrane vitamin D receptor (VDRm) [138]. The phenomenon of rapid calcium flux induced by 1,25(OH)_2_Ds in the intestine, known as transcaltachia, has been extensively studied. This process requires complicated and specific signaling pathways involving voltage-gated L type channels and protein kinase C [139,140,141]. The 1,25(OH)_2_D–VDRm complex also promotes the activation of several intracellular second messengers, controlling the activity of different kinases such as PKA, PKB, and MAPK [142]. Once 1,25(OH)_2_Ds interact with intracellular signaling molecules or transcription factors through VDRm, expression of various genes is influenced, leading to the modulatory effects of 1,25(OH)_2_-D on immunity, antiviral responses, and cell survival. Protein–protein interactions between VDRm and target proteins, such as inhibitor of nuclear factor kappa-B kinase subunit β (IKKβ) [143], RunX1 [144], Stat1 [145], cAMP [146], are involved in this process. IKKβ is one of the upstream regulators of the canonical NF-κB pathway, a transcription factor that regulates various genes, including MMP-9 [147].

### 3.3. Functions of Vitamin D in the Human Body and Related Diseases

Vitamin D plays an important role in maintaining calcium homeostasis. Vitamin D can enhance calcium absorption from the intestines by stimulating the synthesis of calbindin 9K, a calcium binding protein, and by inducing two major calcium transporters, TRPV5 and TRPV6, in the intestinal mucosa [6]. For calcium mobilization from bone, vitamin D and parathyroid hormone (PTH) work synergistically. Both vitamin D and PTH impact osteoblasts by reacting with VDR, leading to an increase in the expression of genes encoding bone matrix proteins like osteocalcin and osteopontin [148]. Both 1,25(OH)_2_Ds and PTH stimulate the synthesis of receptor activator of nuclear factor κ-B (RANK) ligand, which binds to RANK on osteoclasts, promoting their differentiation and activity, leading to bone resorption [5,149]. Additionally, 1,25(OH)_2_Ds and PTH increase calcium reabsorption in the kidney’s distal convoluted tubules, reducing renal calcium excretion.

Moreover, as summarized in Table 3, vitamin D is involved in maintaining various physiological conditions, including: (i) modulating the immune system against infections and the risk of autoimmune diseases and (ii) regulating cell growth, differentiation, and apoptosis, which are essential for maintaining healthy tissues and preventing the development of cancer. Investigation of cell cycle controlling systems has reported vitamin D’s participation in cell cycle arrest through controlling expression of different regulatory molecules such as HIF1a, p53, MYC, Ras, MAPK, BRCA1, and GADD_45_ [150]. Alteration of vitamin D metabolism can be observed in various pathological conditions such as rickets, osteomalacia, renal dystrophy, essential hypertension, multiple sclerosis, rheumatoid arthritis, and different cancers [151].

Vitamin D deficiency is associated with the risk of many other extra-skeletal diseases, including cancers [152,153]. Vitamin D intoxication can increase the degree of saturation sufficiently to increase the free concentrations of 1,25(OH)_2_Ds and so cause hypercalcemia without necessarily raising the total concentrations [154]. The high levels of vitamin D can cause hypercalcemia, which is a condition where the blood calcium levels are too high [155]. Hypercalcemia has various clinical manifestations that affect multiple organ systems [156].

Although vitamin D_2_ and D_3_ share structural similarity, the functional equivalence for human health has been debated in recent years. Multiple studies suggest that they have equal effectiveness in raising circulating serum 25(OH)D concentration, while some other studies provide evidence that vitamin D_3_ is more efficient compared to vitamin D_2_ [157,158,159]. The comparison of their effects showed that vitamin D_2_ is less effective than D_3_ in raising circulating serum 25(OH)D in acute studies [158]. Long-term daily administration studies reported higher efficacy of D_3_ [160,161] or equal efficacy [162]. Some studies have concluded that vitamin D_3_ is more effective than D_2_ in reducing cancer and all-cause mortality, regulating gene expression, and shifting the immune system to a more tolerogenic status [161,163].
ijms-24-17038-t003_Table 3Table 3Physiological functions of vitamin D and related diseases.FunctionActivityRelated DiseaseRef.Calcium homeostasisFacilitation of calcium absorption in intestine and resorption in the renal tubulesRickets[6]Osteomalacia[5,149]Immune regulationRegulation of the expression and activity of pro-inflammatory cytokines (IL-1β, IL-6, IL-8, TNF-α, and IFN-γ)Rheumatoid arthritis[164]Multiple sclerosis[165,166]Allergic diseases[167]Chronic diseases[168]Cardiovascular disease[19,169]Cancer[170]Cell growth regulationRegulation of the expression of several genes involved in proliferation and differentiationCancers[171]Autoimmune diseases[172]MMP regulationDirect and indirect regulation of the expression of MMP-2 and MMP-9Cancers[173,174,175,176,177]Liver fibrosis[178]

### 3.4. Vitamin D in Cancer Treatment and MMP Regulation

Vitamin D has been extensively considered for its potential as an anticancer agent based on epidemiological and preclinical studies. The extensive epidemiologic evidence strongly supports the importance of sufficient vitamin D nutrition, which includes sunlight exposure, in preventing various types of cancers with particular focus on breast, colon, and prostate cancers [179,180,181]. Early reviews of multiple meta-analyses of epidemiological studies have shown that higher vitamin D intake or higher levels of 25(OH)D are associated with a significant reduction in the risk of development of colorectal and breast cancers, especially in premenopausal females [180,182,183]. A long-term epidemiological study on pancreatic cancer patients has also shown that the patients who had sufficient prediagnostic plasma levels of 25(OH)D had longer survival [183]. Similarly, pretreatment serum vitamin D deficiency was associated with increased inflammatory biomarkers in all stages of pancreatic ductal adenocarcinoma [184]. However, the later epidemiological study based on a randomized, double-blind, placebo-controlled trial reported that daily supplementation with vitamin D_3_ (1000 IU), calcium (1200 mg), or both after removal of colorectal adenomas did not significantly reduce the risk of recurrent colorectal adenomas over a period of 3 to 5 years [185]. Furthermore, multiple in vivo and in vitro studies demonstrated vitamin D and its metabolites, especially 1,25(OH)_2_Ds have anticancer effects through various pathways, including genomic and non-genomic pathways [186].

Among the mechanisms, vitamin D and its metabolites could stimulate the expression and activity of MMP-2 and MMP-9. The antitumor effects of 1,25(OH)_2_D, by blocking vasculogenic mimicry (VM) growth factors and altering TIMP/MMP balance in breast cancer cells have been reported [173]. There is also a reduction in the expression and activity of MMP-2 and MMP-9 in breast cancer cells, MCF-7 and MDA-MB-231, upon treatment of 1,25(OH)_2_D. The expression of TIMP-1/2—natural inhibitors of MMPs—in these cell lines was upregulated, while a decrease in VEGF which regulates the expression of MMP-9 was also observed [117,118].

The treatment of ovarian cancer cells with 1,25(OH)_2_D_3_ decreased the expression of MMP-9 in M2 macrophages [174]. This led to the suppression of cell proliferation and migration abilities in ovarian cancer. In a study in human corneal epithelial cells (HUCEC), the expression of MMP-9 was decreased with 1,25(OH)_2_D_3_ treatment [187]. The study demonstrated that HCECs are able to produce 1,25(OH)_2_D_3_ themselves from precursors D_3_ and 25(OH)_2_D_3,_ resulting in an enhanced expression of the antimicrobial peptide, LL-37, dependent on VDR 25(OH)_2_D_3_ decreasing the expression of proinflammatory cytokines (IL-1β, IL-6, TNFα, and CCL20) and MMP-9.

The downregulation of MMP-2 and MMP-9 expression and activity upon addition of 1,25(OH)_2_D and/or 25(OH)D in human lung fibroblasts cells (HFL-1) have been reported as well [175]. This downregulation correlated with the inhibition of IL-1β, an inhibitor of TIMP-1 and TIMP-2. Therefore, vitamin D, 25(OH)D, and 1,25(OH)_2_D play a role in regulating human lung fibroblast functions in wound repair and tissue remodeling. Also, the combination of RA and 1,25(OH)_2_D_3_ has shown potential as a preventive agent against cell invasion in pancreatic adenocarcinoma (PDAC) [188]. The observed anticancer activity in PDAC cells resulting from the treatment with RA and 1,25(OH)_2_D_3_ was linked to the inhibition of MMP-9 expression. The suppression of TNF-α played an essential role in this inhibition, as it effectively blocked the JNK pathway and downregulated miR-221 expression.

Furthermore, the potential of vitamin D as a therapeutic agent for a number of human diseases, besides cancers, have been suggested by altering the expression and activity of MMPs. The expression of MMP-2 and MMP-9 were decreased in response to vitamin D metabolites which could be a VDR therapy to improve arterial calcification [189]. The daily intake of vitamin D may suppress MMP activity and be involved in the development of articular cartilage degeneration and the progression of osteoarthritis [19].

## 4. Vitamin E

Vitamin E was discovered in 1922 by Evans and Bishop [190]. Vitamin E exists in multiple forms, α-tocopherol is the most biologically active form. It is distributed throughout the body and found in various tissues and organs. Vitamin E is not a single nutrient, but a group of compounds that consist of four tocopherol isomers (α-, β-, γ-, and δ-tocopherol) and four tocotrienol isomers (α-, β-, γ-, and δ-tocotrienol), as lipophilic antioxidants preventing lipid peroxidation (Figure 4) [191]. The bioavailability of vitamin E depends on pancreatic function, biliary secretion, micellar formation, and penetration across intestinal membranes [192].

### 4.1. Transportation and Homeostasis of Vitamin E

The distribution of vitamin E takes place throughout the body and its absorption occurs in the intestine, where it is taken up alongside lipids, and packaged into lipoproteins for transportation to various tissues and organs. With the carriers such as chylomicron remnants, low-density lipoproteins (LDLs), and high-density lipoproteins (HDLs), transportation of vitamin E is facilitated [193,194,195]. A substantial portion of the absorbed vitamin E is stored in adipose tissue, with estimates suggesting that approximately 90% of the total amount is deposited, specifically in the lipid droplets of adipocytes [196]. This storage mechanism allows the body to access and utilize vitamin E when needed, contributing to its role in various physiological processes. Vitamin E homologs are primarily transported by very low-density lipoprotein (VLDL) with α-tocopherol specifically recognized and transported by the α-tocopherol transfer protein in the liver [197]. VLDL is a type of lipoprotein that carries vitamin E, triglycerides, and cholesterol from the liver to various tissues in the human body. The liver plays a crucial role as a hub for the distribution of vitamin E homologs throughout the body [197].

The metabolism of vitamin E mainly occurs in the liver (Figure 5). The majority of γ-tocopherol, δ-tocopherol, γ-tocopherol acetate, and δ-tocopherol acetate are metabolized through a process initiated by CYP4F2, leading to the production of 13’-OHs and 13’-COOH metabolites [198]. These metabolites further undergo conversion into the ultimate metabolite, CEHC (carboxyethyl hydroxychroman) [199]. Conversely, most α-tocopherol and small amounts of other vitamin E forms are transported by tocopherol transfer protein (TTP) within hepatic cells, then integrated into lipoproteins with the assistance of ATP-binding cassette transporter A1 (ABCA1) [198,200]. Vitamin E bound to lipoproteins is transported to other tissues through the circulatory system.

The concentration of vitamin E in tissues and organs varies. It is stored mainly in adipose tissue and cell membranes. After reaching the bloodstream, vitamin E is distributed to different tissues, where it serves as an antioxidant protecting cell membranes from oxidative damage. The plasma levels of vitamin E are influenced by the absorption, distribution, and excretion rates of each of its isoforms. All eight homologs possess lipophilic properties and are absorbed from the intestine after being ingested in micelles, which are formed by pancreaticobiliary secretions [201]. In plasma, the half-life of α-tocopherol is estimated to be around 20 h, the longest among all the vitamin E isoforms [202]. Therefore, due to the longest half-life, α-tocopherol is the predominant isoform found in tissues whereas the other congeners are metabolized and more quickly removed [203]. Plasma α-tocopherol concentrations in humans range from 11 to 37 μM whereas γ-tocopherol concentrations are roughly 2 to 5 μM, and other tocotrienol concentrations are less than 1 μM [204]. Vitamin E tends to accumulate in high-lipid tissues including the liver, adipose tissue, muscles, and brain, and protects against oxidative damage. The levels of vitamin E in the bloodstream are regulated by metabolic processes, including its assembly and secretion in lipoproteins in the intestine and liver, transfer between lipoproteins in the blood, and uptake by various tissues. This ensures plasma vitamin E levels are closely linked to normal lipoprotein metabolism in the body [195].

Regulation mechanisms of vitamin E encompass various processes by which it influences cellular and molecular activities in the body. Vitamin E is well-known for its antioxidant properties, but it also plays a role in non-antioxidant functions and regulation of cellular processes. These mechanisms encompass the inhibition of mitogen-activated protein kinase (MAPK) signaling pathways [205,206], modulation of transcription factors, anti-inflammatory effects, and impacts on cellular signaling pathways [207,208,209,210,211]. Collectively, these mechanisms contribute to its role in maintaining cellular health and addressing various health-related conditions.

### 4.2. Functions of Vitamin E in the Human Body and Related Diseases

Under normal conditions, vitamin E is an essential nutrient for a vital role as an antioxidant in the human body [212]. Vitamin E, together with other vitamins and micronutrients, fulfills various physiological roles in maintaining the body’s overall balance and health [213]. Its deficiency can contribute to the development of neurological disorders [214]. The roles and related diseases of vitamin E are summarized in Table 4.

Vitamin E has demonstrated effectiveness in combating conditions such as cancer, aging, arthritis, and cataracts due to its antioxidant properties [215]. Therefore, the antioxidant effect of vitamin E has been attributed to a wide range of benefits, such as anti-inflammatory, anticancer, and neuroprotective effects. In addition, vitamin E has been well-documented to potentially affect endothelial nitric oxide synthase, including vasculoprotective, antifibrotic effects, and wound healing [216,217,218]. The localized administration of vitamin E may offer more favorable outcomes than systemic administration in cancer treatment for humans [216]. Furthermore, the various vitamin E isoforms enact a key role in safeguarding cell membranes, which are abundant in highly unsaturated fatty acids, against oxidative damage with α-tocopherol being the most biologically active and widely recognized for its role in this regard [219].

Moreover, vitamin E could stimulate the defense systems of the human body by enhancing humoral and cell immune responses, and increasing phagocytic functions. Its supplementation significantly strengthens both cell-mediated and humoral immune functions in humans, especially in the elderly [215]. A recent study demonstrated that daily vitamin E supplementation can improve the immune response to a specific antigen [220]. Vitamin E has been shown to act as an antioxidant, modulate signal transduction, regulate gene expression, and play a role in managing skin diseases [221]. Furthermore, topical vitamin E, applied directly to the skin, has emerged as a popular treatment for a number of skin disorders owing to its antioxidant properties. It protects the skin from various deleterious effects due to solar radiation by acting as a free radical scavenger [222].

Cardiovascular complications arise because of the oxidation of LDLs present in the body and the consequent inflammation [223]. By scavenging free radicals and preventing the oxidation of LDLs, γ-tocopherol contributes to the maintenance of vascular health. Additionally, γ-tocopherol has been reported to improve cardiovascular function by increasing the activity of nitric oxide synthase, which produces vessel-relaxing nitric oxide [224]. Moreover, vitamin E could modulate the development of cardiosclerosis and play an important role in cardiovascular disorders, including ischemic heart disease and heart failure [225].

Vitamin E deficiency in humans can lead to a range of health issues, including muscle weakness, vision problems, immune system changes, numbness, difficulty walking, tremors, and poor balance. It is also well-documented to cause ataxia, a neurological disorder resulting from sensory neuron damage in the peripheral nervous system [226,227]. In relation to neurological diseases, long-term α-tocopherol supplementation has been shown to be effective in preventing the progression of nervous system degeneration caused by vitamin E deficiency over decades [228]. In AD, oxidative stress and aggregation of amyloid-β (Aβ) cause neuronal damage and neuronal cell death. The antioxidant properties of vitamin E could reduce oxidative stress, prevent cytotoxic hydrogen peroxide production, and protect neurons, potentially slowing AD progression [215]. Additionally, Vitamin E deficiency may lead to neuromuscular issues like spinocerebellar ataxia and myopathies [229]. Furthermore, vitamin E deficiency can result in anemia due to oxidative damage to red blood cells [229]. It has also been associated with retinopathy [230] and can impair the immune response [231]. Ensuring adequate vitamin E intake is essential for maintaining the overall health of the nervous system and various other systems in the body.
ijms-24-17038-t004_Table 4Table 4Physiological functions of vitamin E and related diseases.FunctionActivityRelated DiseasesRef.AntioxidantMitigates oxidative stress and counteracts free radicalsCancer[232]Aging[232,233]Arthritis[234,235]NeuroprotectionReduction in oxidative stress and aggregation of amyloid-β (Aβ)AD[215,236,237]Parkinson’s disease (PD)[238,239]Cardiovascular healthEnhancement of nitric oxide synthase activityHeart disease[223]Stroke[240,241]Skin healthActing as a free radical scavengerSkin damage[222,242]Skin cancer[221]MMP regulationRegulating the expression of MMP-2 and MMP-9 through specific pathways, indirectlyCancer[207,208,209]Inflammatory disorders[20]

### 4.3. Vitamin E for Treating Human Diseases and MMP Regulation

In diseased conditions, the roles of vitamin E become important. Its antioxidant properties may play a significant role in mitigating oxidative stress and inflammation, which are often key factors in the onset and/or progression and severity of various diseases. Based on multiple epidemiological studies, vitamin E possesses cancer-preventive potential. It exhibits anticancer properties by stimulating the wildtype p53 tumor suppressor gene, downregulating mutant p53 proteins, activating heat shock proteins, and blocking transforming growth factor-α to exert an antiangiogenic effect. These diverse functions contribute to the potential of vitamin E in combating cancerous processes [243].

Investigation of vitamin E and cancer has primarily focused on α-tocopherol. However, other forms of vitamin E, especially γ-tocopherol, may exhibit distinct mechanistic properties that are relevant to the prevention of lung cancer [244]. For example, a case-control study in Europe reported that γ-tocopherol can reduce the risk of lung cancer [192]. Furthermore, γ-tocopherol, especially in combination with δ-tocopherol, induced apoptosis in androgen-sensitive prostate cancer cells within a short duration of three days, while α-tocopherol alone did not exhibit the same effect [245].

Both in vitro and in vivo studies have demonstrated that tocotrienols exhibit more potent anticancer activities compared to tocopherols. Among the tocotrienols, γ-tocotrienol and δ-tocotrienol have been particularly highlighted for displaying stronger anticancer effects by inhibiting cancer invasion and metastasis [246,247,248,249,250,251]. δ-tocotrienol has been reported to inhibit cancer cell invasion by downregulating MMP-2 and MMP-9 [252]. Treatment with γ-tocotrienol also resulted in the suppression of mesenchymal markers and the restoration of epithelial markers, which are associated with the inhibition of cell invasion [253]. Moreover, recent findings indicate that tocotrienols exert an impact on numerous signaling pathways within cancer cells, including NF-κB-mediated pathways, phosphatidylinositol-3 kinase/phosphoinositide-dependent/Akt, Raf/Erk, and JNK-related pathways [207,208,209,210]. Tocotrienols have shown impressive anticancer properties over time, consistently surpassing tocopherols in their ability to combat tumors [247,251], while also preserving normal cell growth and viability [211,254].

Vitamin E indeed exerts a wide range of effects beyond its potential in the treatment of cancer and it has been reported to have neuroprotective roles in maintaining cognitive function and reducing the risk of neurodegenerative diseases. This is attributed to its remarkable antioxidant, anti-inflammatory, and cholesterol-lowering properties [255]. Due to its antioxidant properties, vitamin E has emerged as an appealing therapeutic agent for the prevention and treatment of neurodegenerative diseases including AD and PD where oxidative stress is one of the important pathophysiological risk factors [238,256]. Therefore, vitamin E supplementation has been extensively studied as a potential therapy for neurodegenerative disorders both in vitro and in vivo [238,239,257].

Extensive evidence supports the involvement of ROS as crucial pathologic mediators in numerous human disease processes. Specifically, O_2_^−^ radicals actively participate in regulating the activities of MMPs. Vitamin C and E act as potent inhibitors of oxygen-free radicals [241], which are known activators of MMPs. They have an important role in preventing the excessive activity of MMPs which can be linked to cancer progression and metastasis. Therefore, these vitamins could be beneficial in the prevention and management of certain diseases and pathological conditions associated with MMP dysregulation [258,259,260]. Also, it has been reported that α-glucosylation (AGR) as well as vitamins C and E have the ability to decrease the expression and activity of MMPs.

In addition, oral supplementation with vitamin E in diabetic and obese mice led to a reduction in oxidative stress levels along with a decrease in the expression of MMP-2 and improved skin tensile strength and collagen fibers [261]. Chronic inflammation can lead to increased MMP production and activity, contributing to tissue damage and disease progression. By reducing inflammation, vitamin E may help modulate MMP-related processes. The administration of vitamin E supplements to patients with diabetic nephropathy resulted in reduced circulating levels of TNF-α, MMP-2, and MMP-9 compared to the placebo group. After 12 weeks of intervention, compared with the placebo, vitamin E supplementation resulted in a significant reduction in MMP-2 and MMP-9 [20].

α-tocopheryl succinate, a derivative of vitamin E, effectively inhibits the invasion of human prostate cancer cells, including PC-3 and DU-145, through a reduction in secreted MMP-9 activity [262]. The treatment of Kirsten murine sarcoma virus-transformed rat kidney (KNRK) cells with vitamin E, led to the inhibition of MMP-9 and MMP-2 activities [263]. Based on the inhibitory effects on NF-κB and STAT3, γ-tocotrienol and δ-tocotrienol have been proposed to be useful in chemoprevention or adjuvant chemotherapy for cancer. The treatment with γ-tocotrienol or its combination with gemcitabine resulted in the downregulation of NF-κB-regulated gene products, including cyclin D1, MMP-9, and CXCR4 [264].

Furthermore, vitamin E can regulate vital cellular processes, including inflammation and cell proliferation, by modulating key signaling pathways such as MAPKs (ERK, JNK, and p38) and NF-κB. The MAPKs and the transcription factor NF-κB play important roles in regulating the expression of MMP-2 and MMP-9 [265,266]. The activation of MAPKs and NF-κB could upregulate the levels of MMP-2 and MMP-9 in cancer cells and other pathological conditions. The treatment of p38 mitogen-activated protein kinase (p38-MAPK) inhibitor (SB203580), p38-MAPK siRNA, or NF-κB inhibitors (TPCK and PDTC) could suppress the expression and promoter activity of MMP-2 as well as cell invasion [267]. Consequently, by inhibiting these pathways, vitamin E will indirectly suppress the activity and expression of MMPs.

## 5. Vitamin K

Vitamin K plays a multitude of vital roles in the body despite its relative lack of popularity compared to other essential vitamins [268]. Naturally, vitamin K, as presented in Figure 6, exists in three different forms: K_1_ (phylloquinone), K_2_ (menaquinones), and K_3_ (menadione) [269]. Vitamin K_1_ (phylloquinone) is abundant in green leafy vegetables and is an essential dietary source of vitamin K. Vitamin K_2_ is synthesized by the human intestinal microbiota and the primary sources of vitamin K_2_ are of microbial origin, commonly found in fermented foods such as cheese, curds, and animal livers [270,271]. In contrast, vitamin K_3_ (menadione) is produced through chemical synthesis and is primarily used for industrial and research purposes, rather than occurring naturally [272].

### 5.1. Transportation and Homeostasis of Vitamin K in Human Body

Vitamin Ks are absorbed alongside dietary fats, and the presence of bile salts enhances their bioavailability (Figure 7) [273]. After being absorbed in the intestines, both vitamins K_1_ and K_2_ enter through the bloodstream as a component of chylomicrons, which are large lipoprotein particles formed in the intestines after dietary fat absorption. Once vitamin Ks are in the bloodstream, chylomicrons are taken up by the liver [274,275]. In the liver, vitamin K_1_ undergoes metabolism and over half of the absorbed amount is excreted by the human body. Vitamin K_2_ is carried by LDL from the intestine to tissues outside the liver for various physiological functions [276,277]. Vitamin K_2_ exhibits a preference for accumulation in peripheral tissues. It is notably found in high levels in the brain, aorta, pancreas, and adipose tissue indicating its vital roles in these areas [278,279,280]. In contrast, the liver contains relatively lower levels of vitamin K_2_. This selective distribution highlights the specific importance of vitamin K_2_ in various extrahepatic functions and tissues [274,275]. On the other hand, vitamin K_3_ is not transported in the same way as vitamin K_1_ and K_2_. Thus, it is commonly administered through supplements or injections for specific therapeutic or research purposes.

Vitamin K_1_ has a relatively fast removal rate from the circulation based on the monitoring of both urine and bile [10,281,282]. It is primarily retained in the liver to facilitate the carboxylation of clotting factors [277]. The ability of the liver to store and utilize vitamin K_1_ for this purpose is essential for maintaining a balanced and effective blood clotting mechanism in the body [283]. Conversely, vitamin K_2_, especially its long-chain derivatives, undergoes redistribution back into the circulation instead of being primarily retained in the liver. This redistribution enables its availability for tissues beyond the liver. This includes important tissues such as bones and vasculature, where vitamin K_2_ plays essential roles in supporting bone health and promoting cardiovascular health [277,284]. This indicates that vitamin K_2_ is not solely confined to the liver but is also utilized by extra hepatic tissues, contributing to its diverse physiological functions in various parts of the human body.

The metabolism of vitamin K in liver is of utmost importance for blood clotting and the regulation of calcium balance within the body as shown in Figure 7. Vitamin K is recycled in the liver after participating in the carboxylation process [285]. After carboxylation, vitamin K becomes an epoxide form, either vitamin K oxide or vitamin K epoxide and must undergo a chemical reduction process to return to its reduced form by the enzyme, vitamin K epoxide reductase (VKOR) [286]. This process allows vitamin K to participate in carboxylation, enabling clotting proteins like factor II, VII, IX, and X to function in blood clotting [10]. Additionally, clotting proteins are further transformed into their active forms by the enzyme γ-glutamyl carboxylase (GGCX) after vitamin K-dependent carboxylation is over [287,288]. Thus, in this metabolic process, clotting proteins are well-equipped to carry out their functions effectively in the liver for maintaining calcium homeostasis.

Vitamin K_1_ and MK-4 (menaquinone-4), a subtype of vitamin K_2_ presented in Figure 6, have different distributions within tissues. Vitamin K_1_ is predominantly concentrated in the liver, heart, and pancreas, while MK-4 is more abundant in the kidneys and brain. Additionally, longer forms of vitamin K_2_ (MK-7 to MK-13) are also detected in the liver, contributing to the diverse physiological functions of vitamin K in the body [289]. This diversity in tissue distribution of vitamin K_1_ and MK-4 underscores their importance in maintaining overall health and highlights the complexity of vitamin K’s role in the human body.

### 5.2. Functions of Vitamin K in the Human Body and Related Diseases

Under normal conditions, vitamin K plays several crucial roles in maintaining overall health and supporting various physiological processes in the human body (Table 5) [288]. Recently, extensive research has revealed that vitamin K offers health benefits in maintaining blood homeostasis which has been linked to chronic low-grade inflammatory diseases, including cardiovascular disease, osteoarthritis, dementia, cognitive impairment, mobility disability, and frailty [290]. Vitamin K is widely utilized in the treatment of various diseases, such as vascular calcification, osteoporosis, diabetes, and liver cancer, because of its procoagulant properties as well as its anti-inflammatory and antioxidant capabilities [291,292,293].

Additionally, vitamin K is necessary for wound healing in animal models, as it could enhance cell proliferation and differentiation [290]. Vitamin K has been shown to reduce the inflammatory response in in vitro, animal [294], and in large-scale human studies [295]. Another crucial aspect of the importance of vitamin K is its ability to function as a potent antioxidant. By generating vitamin K–hydroquinone, a highly efficient radical scavenging complex, this complex effectively mitigates lipid peroxidation in cells [296]. Despite its vital role, vitamin K is not as widely discussed as other vitamins (i.e., A, B, C, D, and E). However, vitamin K is indeed essential for blood clotting and maintaining healthy bone tissue [297]. Its deficiency results in blood coagulation impairment, hemorrhagic disorder, fat malabsorption, and deterioration of bone density [272].

Vitamin K, especially vitamin K_2_, is believed to contribute significantly to cardiovascular health by playing a crucial role in regulating calcium homeostasis, which is vital for maintaining optimal heart function and vascular health. The primary reason for the speculation about the positive cardiovascular effects of vitamin K is its crucial role in the synthesis of Matrix Gla protein (MGP), which acts as a natural inhibitor of arterial calcification, thereby helping to maintain the health of arteries [10]. Vitamin K also plays a crucial role in modifying specific glutamic acid residues in proteins, both inside and outside of the liver through post-translational processes. This process is critical for blood coagulation and preventing calcification in cartilage and blood vessels [298]. Various types of vitamin K exhibit differences in their biological activities, which arise from variations in enzyme affinity and tissue distribution [299]. These distinctions could influence the determination of the specific functions and effects of each form of vitamin K in human body.

A substantial body of evidence establishes a connection between vitamin K deficiency and heightened risks of cancer, cardiovascular disease, soft tissue calcification, and osteoporosis [300,301,302]. Vitamin K deficiency can lead to various health issues, including increased bleeding tendencies, impaired bone development, heightened risk of osteoporosis and fractures, and elevated susceptibility to cardiovascular diseases characterized by vascular calcification and atherosclerotic plaques [303]. Vitamin K deficiency also affects calcium homeostasis, which leads to vascular calcification and bone disorders [15].
ijms-24-17038-t005_Table 5Table 5Physiological functions of vitamin K and related diseases.FunctionActivityRelated DiseasesRef.Blood clottingThe accumulation of calcium in arteries and blood vesselsSynthesis of Matrix Gla protein (MGP) Activation of blood clotting factorsArterial calcification[304,305]Cardiovascular disease[298]Liver disease[306,307]Blood coagulation impairment[308]NeuroprotectionSphingolipid metabolism modulation and Aβ clearanceRegulation of Gas6 carboxylation and neuronal apoptosisAlzheimer’s disease (AD)[309]Neurological disorders[310]Bone healthFostering calcium absorption into boneOsteoporosis[15]Fractures and weak bone structure[311,312]Wound healingEnhanced cell proliferation and tissue repairDiabetesImmune disorders[313][313,314,315]MMPs regulationIndirect influences on MMP-2 and MMP-9 expressionCancer[316,317]Inflammatory disorders[318,319]Cardiovascular disease[320]

### 5.3. Vitamin K in Human Disease Treatment and MMP Regulation

Within the realm of diseased conditions, vitamin K demonstrates its diverse significance and impacts on blood clotting disorders, bone health, cardiovascular disease, liver disease, malabsorption disorders, warfarin therapy, and its potential role in cancer. Vitamin Ks exhibit promising potential as anticancer agents and function as chemosensitizers when combined with other chemotherapy drugs targeting diverse cancers from various origins through multiple mechanisms, such as inhibition of cancer cell proliferation, survival, metastasis, and angiogenesis, and induction of intrinsic and extrinsic apoptosis, nonapoptotic cell death, autophagy, cell cycle arrest via inhibition of MAPK/ERK, NF-κB, wingless-related integration site (WNT), JNK, and phosphatidylinositol 3-kinase/protein kinase B (PI3K/AKT) signaling pathways [321].

Vitamin Ks have been investigated as therapeutic agents for cancer treatment of various cancer cell lines [322], including HCC [323], leukemia, colorectal cancer, ovarian cancer, pancreatic cancer [324], and lung cancer [325]. In different cancer cell lines, vitamin K_2_ could inhibit cancer cell growth through the induction of autophagy, a natural process that eliminates damaged cellular components and safeguards against diseases [326]. Furthermore, vitamin K_2_ could inhibit the growth of hepatocellular carcinoma cells by suppressing cyclin D1 expression through inhibition of NF-κB activation [327]. Vitamin K_3_ has been reported to inhibit the proliferation of several cancer cells as cellular redox mediators, producing ROS, and triggering apoptosis through mitochondrial pathways [328,329]. Vitamin K_3_ treatment, either alone or in combination with other chemotherapeutic drugs, has demonstrated the ability to inhibit the growth of various neoplasms (abnormal growths of tissue) and sensitize drug-resistant cancer cells to standard chemotherapy [321].

The combined action of vitamin K_3_ and vitamin C (also known as vitamin K_3_/AA) has been reported to demonstrate a synergistic effect in inducing cell death in various cancer types. When used together, they work in tandem to enhance the anticancer properties. It was reported that K_3_/AA treatment in a 1:100 ratio potentiated the antitumor effect by about 4- to 61-fold in urologic cancer cells even with a short incubation time such as 1 h [330]. The later in vivo study acknowledged that the combination of vitamin K_3_ and vitamin C also decreased activities of plasma MMP-2/-9 in C57BL/6 mice [331]. In a recent study, expression of Siah2 and HIF-1α and MMP-9 were downregulated in colon tissues with the treatment of vitamin K_3_ [332]. Vitamin K and vitamin C alone or in combination, induce apoptosis in leukemia cells by a sequential cascade of molecular events involving the production of ROS, simultaneous activation of NF-κB/p53/c-Jun transcription factors, mitochondrial depolarization, and the caspase-3 activation pathway [333]. However, vitamin K_3_ could exhibit toxic effects on certain cells at high concentrations. Vitamin K_3_ showed significant cytotoxicity against human oral tumor cell lines (HSC-2, HSG), human promyelocytic leukemia cell line (HL-60), and human gingival fibroblasts (HGFs) [334]. Thus, its use as a potential cancer treatment requires careful consideration and proper dosage to minimize harm to healthy cells. The combination of K_3_/AA are harmless to lymphocytes, at least under the present in vitro conditions [333]. In addition, the combination of vitamin K_3_ and D-fraction (DF) could lead to a drastic >90% viability reduction. In addition, they could induce a profound reduction in ACHN cell (Human Renal Adenocarcinoma Cells) viability, through a p21(WAF1)-mediated G1 cell cycle arrest, and ultimately induce apoptosis [335].

Moreover, the potential role of vitamin K in supporting brain health and cognitive function in AD pathology has been studied for a long time. The involvement of vitamin K in brain physiology occurs via the carboxylation of Gas6, a vitamin K-dependent protein, potentially providing protection against neuronal apoptosis triggered by oxidative stress Aβ accumulation [309]. Vitamin K has demonstrated its influence on the onset and progression of AD along with cognitive functions by enhancing Aβ clearance through the modulation of sphingolipid metabolism [336], exerting positive effects on the underlying mechanisms involved in the pathology of AD [310]. Furthermore, a deficiency in vitamin K has been associated with brain aging and cognitive decline, especially in individuals with AD and the elderly [337]. While the research indicates a potential link between vitamin K and AD, it is essential to acknowledge that further extensive research is needed to definitively establish the relationship between vitamin K and the risk of AD.

Beyond that, in particular, vitamin K_2_ is believed to contribute to cardiovascular health by regulating calcium homeostasis, modifying systemic calcification, and reducing arterial stiffness [320]. It could prevent the accumulation of calcium in arteries and blood vessels, supporting vascular health and reducing the risk of atherosclerosis [320]. The positive cardiovascular effects of vitamin K are primarily attributed to its role in the synthesis and activation of Matrix Gla protein (MGP). By blocking the calcification of arteries, MGP helps maintain the flexibility and integrity of blood vessels and reduces the risk of arterial stiffness [10]. The close association between vitamin K and arterial health solidified its importance in fostering a healthy cardiovascular system.

Emerging evidence suggests that vitamin K, particularly vitamin K_2_, may have broader functions beyond blood clotting and bone health, and inhibitory effects on certain MMPs. In a recent study, different forms of vitamin K_2_ (MK-4, MK-5, MK-6, MK-7) inhibited the expression of MMP-2 and MMP-9 in the murine macrophage cell line (RAW 264.7) [319]. As with many biological processes, the regulation of MMPs by vitamin K is complex and involves interactions with various cellular pathways. Vitamin K could induce apoptosis through different biochemical pathways, including alteration of intracellular calcium homeostasis and activation of the following pro-apoptotic factors: JNKs, Fas-dependent and Fas-independent pathways, and NF-κB [316,317]. The potential of vitamin K to inhibit certain pathways suggests it may have a role in further inhibiting MMP activity.

Inhibitors targeting ERK and JNK effectively suppressed 12-O-tetradecanoylphorbol-13-acetate (TPA)-induced AP-1 transcriptional activity, whereas the p38 inhibitor showed no effect in this context. Similarly, Vitamin K_2_ also showed a suppressive effect on TPA-induced MMP expression by reducing AP-1 activity [21]. NF-κB activity is necessary for the upregulation of several MMPs, even in cases where the promoter regions of certain MMPs do not seem to have a clear NF-κB binding site [338]. Vitamin K_2_ could suppress the expression of MMPs that possess NF-κB binding motifs in their promoter regions [21]. Additionally, vitamin K_2_ has been found to enhance the inhibitory effect of 5-fluorouracil on the growth of hepatocellular carcinoma (HCC) cells by inhibiting the activation of NF-κB [339]. Therefore, the vitamin K2-mediated inhibition of NF-κB is thought to be involved in suppressing multiple MMPs.

## 6. Conclusions

Based on the literature included in this review, it appears that fat-soluble vitamins could ameliorate human diseases through various mechanistic pathways, with particular emphasis on MMP-2 and MMP-9 and their related intracellular signals. They could regulate the production of MMP-2 and MMP-9 through various pathways (Figure 8). Vitamin A and D appear to primarily regulate the transcription of MMP-2 and MMP-9 through their respective metabolites, either directly or indirectly. Through the direct pathway, the metabolites of both vitamins can directly regulate the promoter region of the target genes by activating their respective receptors (RARs and RXRs for Vitamin A, and VDRn for Vitamin D). Meanwhile, the indirect pathway involves the activation of receptors like the putative plasma membrane vitamin D receptor (VDRm), which interacts with various intracellular signals to influence the expression of MMPs. Those intracellular signals include (IKKβ), NF-κB, VEGF, interleukin-1 (IL-1), P38-MAPK, TNF- α and JNK. Unlike vitamins A and D, few studies cover the involvement of vitamins E and K in regulation of MMPs by affecting intracellular signaling pathways, such as P38-MAPK, TNF-α, JNK, NF-κB, VEGF, and IL.

Although the fat-soluble vitamins showed their potentials as treatments for human diseases (including cancers) in various in vitro experiments, more clinical and preclinical trials should be performed to determine the proper dose and strategy of consumption. In addition, the delivery mechanisms of vitamins in the human body by lipids, lipoproteins, and/or vitamin binding proteins needs further investigation. With the improvement in understanding the roles of vitamins in pathologies of human diseases through more clinical trials and studies, people can take vitamins in an efficient way as medications and develop new derivatives of vitamins as promising drugs.

In summary, fat-soluble vitamins (vitamin A, D, E, and K) have undergone extensive investigation for their potential applications in treating various human diseases, such as cancers. Specifically, given the involvement of MMPs in the pathogenesis of these diseases, there has been notable research into utilizing fat-soluble vitamins to target MMPs, both directly and indirectly, particularly for regulating the expression and activity of MMP-2 and MMP-9. This review has provided an overview of current studies and knowledge concerning the use of fat-soluble vitamins in targeting MMPs, revealing their potential for the treatment of human disease.

## Figures and Tables

**Figure 1 ijms-24-17038-f001:**
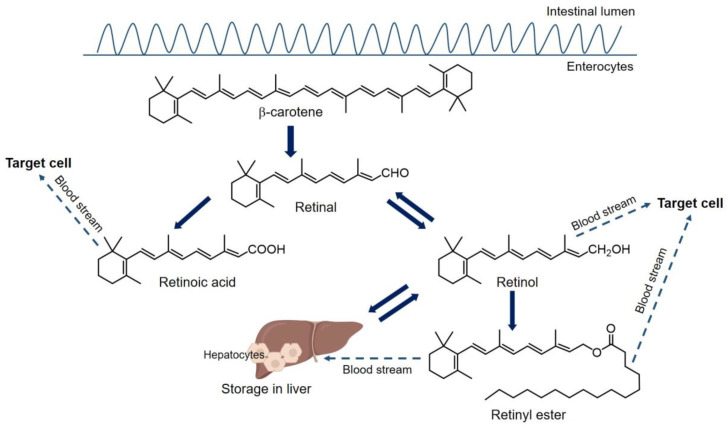
Metabolism and transportation pathways of vitamin A in the human body.

**Figure 2 ijms-24-17038-f002:**
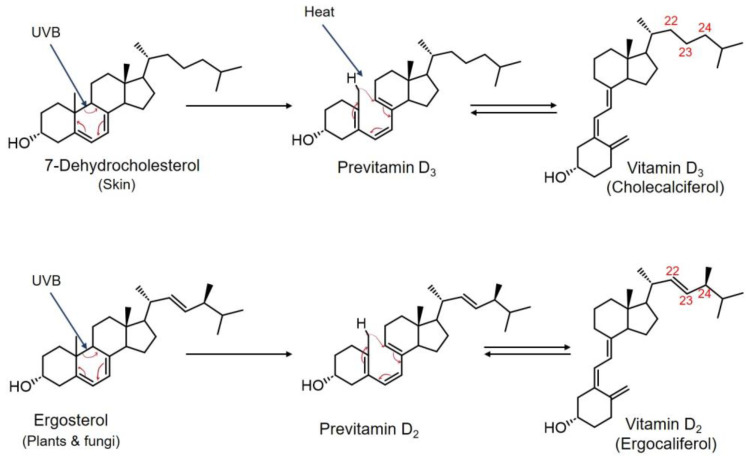
Structure and production mechanism of vitamin Ds.

**Figure 3 ijms-24-17038-f003:**
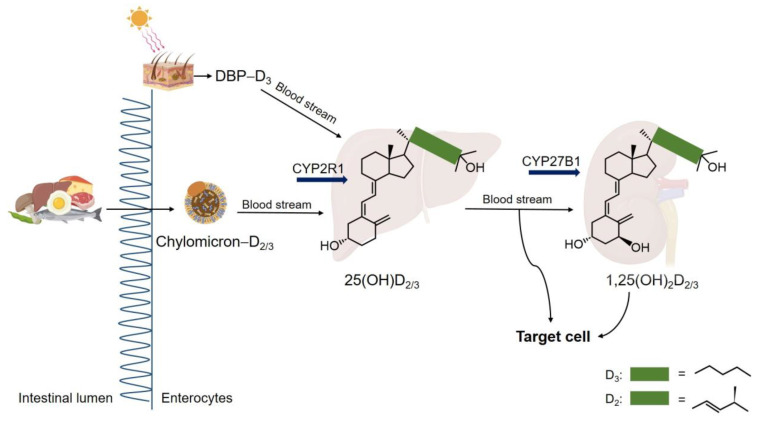
Metabolism and transportation pathways of vitamin D.

**Figure 4 ijms-24-17038-f004:**
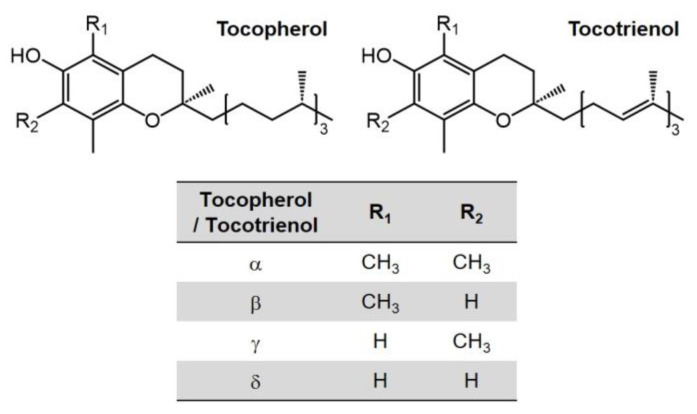
Chemical structures of multiple forms of vitamin E.

**Figure 5 ijms-24-17038-f005:**
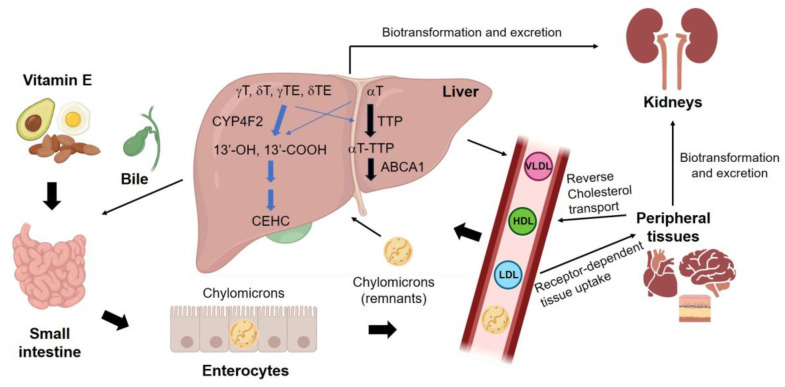
The transportation and metabolism of vitamin E. γT, γ-tocopherol; δT, δ-tocopherol; γTE, γ-tocotrienol; δTE, δ-tocotrienol.

**Figure 6 ijms-24-17038-f006:**
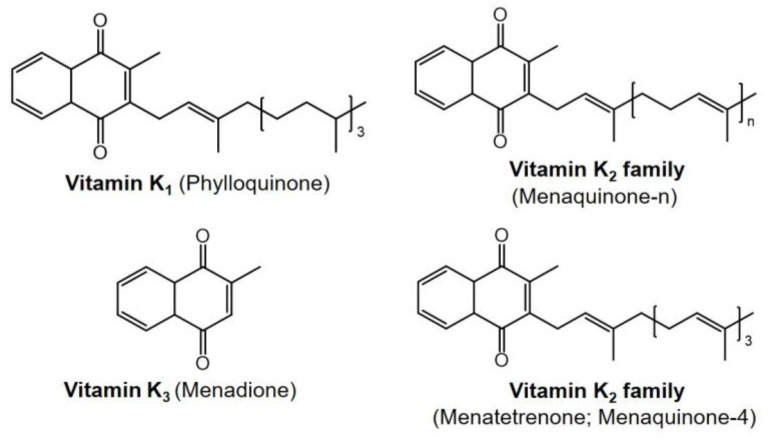
Chemical structures of vitamin Ks.

**Figure 7 ijms-24-17038-f007:**
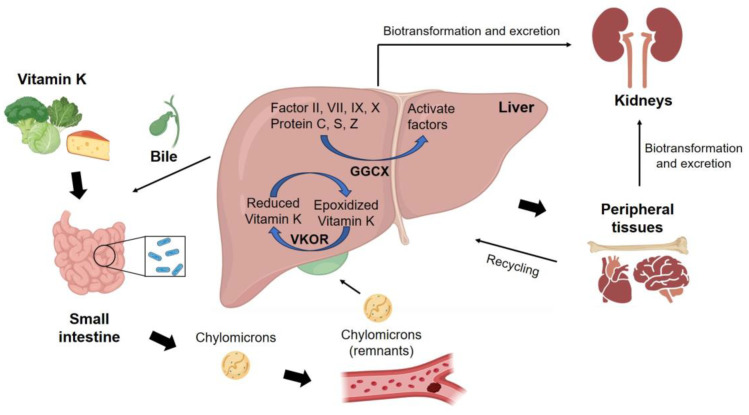
Metabolism and transportation pathways of vitamin K.

**Figure 8 ijms-24-17038-f008:**
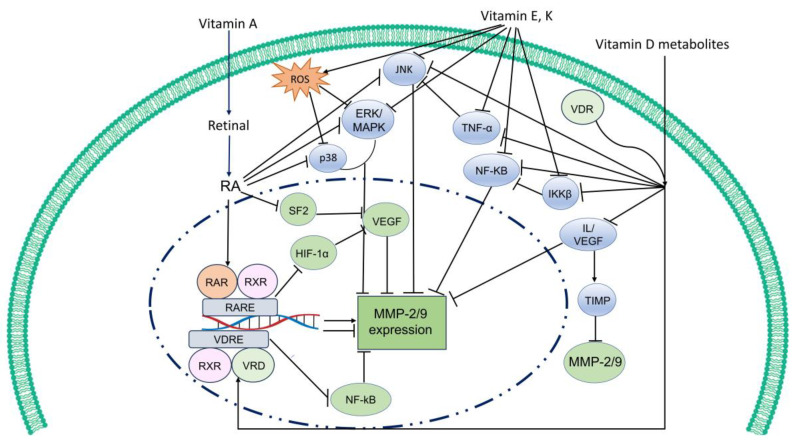
The effect of fat-soluble vitamins on signaling pathways generating MMP-2 and MMP-9.

**Table 2 ijms-24-17038-t002:** Physiological functions of vitamin A and related diseases.

Function	Activity	Related Disease	Ref.
Vision regulation	Rhodopsin generation	Night blindness	[86]
Cell growth and development	Regulation of gene expression	Infectious diseases	[87]
Immune regulation	Promotion of the growth and differentiation of cells for tissue repair Promotion of immune cell differentiation	Acute promyelocytic leukemia	[88,89,90]
		Osteoporosis	[91,92]
Cellular communication	Inhibiting the production of proinflammatory cytokines Protection of immune cells from oxidative stress	Cancer	[17,18,93,94,95]
		Obesity and insulin resistance	[96,97]
Reproduction	Assist the growth and maturation of follicles Regulation of the genes related to spermatogenesis	Congenital malformations	[98]
		Diabetes mellitus and gestational diabetes	[99]
MMP regulation	Regulation of MMP-2 and MMP-9 expression	Cancers	[11,17,18,94,95,100,101]

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
