# Peer review of "Vitamin A, D, E, and K as Matrix Metalloproteinase-2/9 Regulators That Affect Expression and Enzymatic Activity"

_ijms, 2023, doi:10.3390/ijms242317038_

Round 1

Reviewer 1 Report

Comments and Suggestions for Authors

This manuscript summarizes the functions and roles of fat-soluble vitamins in human body under healthy and diseased conditions and their potentials as promising candidates capable of effectively modulating MMPs through multiple pathways. This manuscript is a carefully done overview on the current studies and knowledge concerning the use of fat-soluble vitamins in targeting MMPs, revealing their potential for the treatment of human diseases, which is beneficial for readers in related fields. There are several details should be notice. It is this reviewer’s opinion that the review requires minor revision before being acceptable for publication.

The specific comments are listed in detail as follows:

1. For Figure 6, the “gT, δ-tocopherol; dT, δ-tocopherol; gTE, δ-tocotrienol; dTE, δ-tocotrienol.” in the legend is inconsistent with the “γT, δT, γTE, δTE, and αT” in the liver of Figure 6. What does the “gT, δ-tocopherol; dT, δ-tocopherol; gTE, δ-tocotrienol; dTE, δ-tocotrienol.” mean?

2. Table 1. Expression and activation of MMP-2/9 in different cell types. As with other tables, references need to be added to Table 1.

3. Current research highlights fat-soluble vitamins as potential therapeutics for human diseases, especially cancer. What challenges remain in the transition of these vitamins from potential therapeutic agents to therapeutic agents for clinical use? It is suggested that the authors address these challenges in the conclusion.

Reviewer 2 Report

Comments and Suggestions for Authors

Manuscript contains high amount of too long sentences. Also, there are minor grammatic errors, e.g. line 38, page 1 (cytsokine)

The topic of review is not novel. The issue of fat-soluable vitamins in the context of huamn health and diseases, including associated with MMP-2/9, is known.

The Introduction section is very too long. A lot of data presenting in this section is widely and well known. Also, data about MMP-2 and -9 is not novel. It causes that the manusctipt is boring.

Please indicate the sources of Figures.

It would be valuable to cite other articles describing the role of fat-soluable vitamins in diseases publicated in journals of MDPI, e.g. doi: 10.3390/ijms23084256; 10.3390/nu14112353

Comments on the Quality of English Language

Minor English correction

Reviewer 3 Report

Comments and Suggestions for Authors

General observations:

-       The text below Table 2 does not have the same line spacing as the rest of the text (lines 222-227).

-       Figures 4, 6 and 8 should be improved in terms of background or saved as PNG to remove the black border around the images.

-       Some paragraphs have greater spacing than others, even within the same paragraph.

-       References are not consistent as required by the journal template (e.g. there are Year Year Year).

It has been mentioned that there are few studies on the involvement of vitamins E and K in the regulation of MMPs. In section 5.3, MMP1, 3 and 7 are discussed, whereas the title of the entire review already focuses on MMP2 and 9. This inconsistency should be addressed.

doi: 10.1080/01635581.2011.597537.

doi: 10.3390/cells10071571.

doi: 10.1016/j.biopha.2022.113562.

Round 2

Reviewer 2 Report

Comments and Suggestions for Authors

Authors revised manuscript according to all advices.